# CDK4, CDK6/cyclin-D1 Complex Inhibition and Radiotherapy for Cancer Control: A Role for Autophagy

**DOI:** 10.3390/ijms22168391

**Published:** 2021-08-04

**Authors:** Valerio Nardone, Marcella Barbarino, Antonio Angrisani, Pierpaolo Correale, Pierpaolo Pastina, Salvatore Cappabianca, Alfonso Reginelli, Luciano Mutti, Clelia Miracco, Rocco Giannicola, Antonio Giordano, Luigi Pirtoli

**Affiliations:** 1Department of Precision Medicine, University of Campania “L. Vanvitelli”, 80138 Naples, Italy; antonioangrisani@gmail.com (A.A.); salvatore.cappabianca@unicampania.it (S.C.); alfonso.reginelli@unicampania.it (A.R.); 2Department of Medical Biotechnologies, University of Siena, 53100 Siena, Italy; marcella.barbarino@unisi.it (M.B.); president@shro.com (A.G.); 3Medical Oncology Unit, Grand Metropolitan Hospital “Bianchi-Melacrino-Morelli”, 89124 Reggio Calabria, Italy; pierpaolo.correale@ospedalerc.it (P.C.); roccogiannicola@gmail.com (R.G.); 4Sbarro Institute for Cancer Research and Molecular Medicine, Center for Biotechnology, College of Science and Technology, Temple University, Philadelphia, PA 19104, USA; Luciano.mutti@temple.edu (L.M.); luigipirtoli@gmail.com (L.P.); 5Section of Radiation Oncology, Medical School, University of Siena, 53100 Siena, Italy; pastina.pierpaolo85@gmail.com; 6Pathological Anatomy Unit, Department of Medical, Surgical and Neurological Science, University of Siena, 53100 Siena, Italy; clelia.miracco@unisi.it

**Keywords:** cyclin inhibitors, radiotherapy, autophagy

## Abstract

The expanding clinical application of CDK4- and CDK6-inhibiting drugs in the managements of breast cancer has raised a great interest in testing these drugs in other neoplasms. The potential of combining these drugs with other therapeutic approaches seems to be an interesting work-ground to explore. Even though a potential integration of CDK4 and CDK6 inhibitors with radiotherapy (RT) has been hypothesized, this kind of approach has not been sufficiently pursued, neither in preclinical nor in clinical studies. Similarly, the most recent discoveries focusing on autophagy, as a possible target pathway able to enhance the antitumor efficacy of CDK4 and CDK6 inhibitors is promising but needs more investigations. The aim of this review is to discuss the recent literature on the field in order to infer a rational combination strategy including cyclin-D1/CDK4-CDK6 inhibitors, RT, and/or other anticancer agents targeting G1-S phase cell cycle transition.

## 1. Introduction

The expanding clinical application of CDK4- and CDK6-inhibiting drugs in the management of hormone-sensitive breast cancer [1,2,3,4] has raised a great interest in also testing these G1/S cell-cycle blocking agents in other neoplasms [5,6,7], for which the potential clinical benefit is still largely unknown.

The innovative use of CDK4 and CDK6 inhibitors in combination with hormone therapy (HT) as a treatment for estrogen receptor positive/human epidermal growth factor receptor-2 negative (ESR+/HER2−) metastatic breast cancer (BC) patientsis safe and successful in term of response rate (RR), progression-free survival (PFS), and overall survival (OS). Nevertheless, 10% of patients are refractory to this treatment whereas half of the patients show progression within 24 months of therapy [8]. Therefore, to fully unleash their impact, combining these drugs with other therapeutic approaches seems a rational work-ground to explore.

Even though a potential integration of CDK4 and CDK6 inhibitors with radiotherapy (RT) has been hypothesized, this kind of approach has not been sufficiently pursued, neither in preclinical nor in clinical studies. Similarly, the most recent discoveries focusing on autophagy as a possible target pathway able to enhance the antitumor efficacy of CDK4 and CDK6 inhibitors are promising but need more investigation.

Therefore, our aim is to review and discuss the recent literature in the field in order to infer a rational combination strategy including Cyclin-D1/CDK4 and CDK6 inhibitors, RT, and/or other anticancer agents targeting G1-S phase cell cycle transition.

## 2. Molecular Background and Involved Pathways

The cell cycle has been defined as a set of subsequent events involved in the process of cell replication aiming to pass down the genetic information. In particular, cells are engaged into DNA replication (phase G1) followed by genome replication (DNA synthesis or S-phase), eventually leading to their segregation into two daughter cells (mitosis or M-phase). In a normal cell, this process is finely regulated and cell cycle checkpoints take place entering both S- and M-phase, respectively designated as G1 (between M and S) followed by G2 phase (between S and M). Cell cycle checkpoints represent a pivotal mechanism to preserve cell genomic integrity, prevent uncontrolled proliferation, and allow repair of damaged DNA [9].

Growth factors may induce the activation of positive cell checkpoints that in turn activate and promote cell cycle progression finely tuned by two groups of proteins, designated as cyclins and cyclin-dependent kinases (CDK). At present, four phase-specific cyclins (A, B, D, E) have been described. Cyclin D expression rises in G1 and falls in M phase while cyclin E, cyclin A, and cyclin B expression peaks within G1/S, G2, and G2/M phase, respectively. Experimental evidence suggests that the level of cyclins depends on their degradation by the endoplasmic ubiquitin/proteasome system (UPS).

In order to exert their regulatory functions, cyclins bind specific effector CDKs forming a cyclin/CDK hetero-dimeric complex (CCDK) whose phosphorylation in turn promotes cell cycle progression. Phosphorylation appears to be critical for CCDK functions and is under strict regulation by cyclin-dependent kinase inhibitors (CDKI). At present, two classes of CDKI have been identified: the INK4 family (p15, p16, p18, and p19) and the Cip/Kip protein family (p21, p27, p57) that disrupt the cyclin D1/CDK4 and CDK6 complex and both cyclin A,B/CDK 1 cyclin A,E/CDK 2 complexes, respectively.

CCDK regulation is pivotal in the event of DNA damage or misaligned chromosome during mitosis when negative checkpoint regulators, such as the retinoblastoma family of proteins (Rb), p53, or specific CDKIs, halt the cell cycle and activate DNA repair. In the event of irreparable damage, p53 triggers a pro-apoptotic cascade with consequent cell death [10,11,12,13,14,15]. In transformed cells, either negative or positive checkpoints are deregulated and affect the normal progression of cell cyle. In cancer cells, the additional amplification of the *CCND1* gene, encoding for cyclin D1 (CCND1), is one of the most frequent alterations recorded in human solid neoplasms. This genomic alteration results in (1) *CCND1* overexpression, (2) hyperactivation of cyclin D/CDK4 and CDK6 complex [16], (3) induced expression of E2F target genes, and 4) cell cycle engagement and progression from G1 to S phase.

### 2.1. Antitumor Activity of Cyclin-D1/CDK4 and CDK6 Inhibition

A normal process of cell differentiation is associated with permanent irreversible proliferative quiescence referred to as G0 [17]. In contrast, stem cells involved in tissue renewal and repair (usually in a reversible quiescence) may restart cell cycle progression when stimulated by specific mitogenic factors. Cell cycle engagement and progression (G1/S phase) are regulated by D-type cyclins (D1, D2, and D3) in the early G1, with an outcome depending on the dynamic equilibrium among synthesis, accumulation, localization, and degradation of cyclin D. It has also been described that CCND1/CDK4 and CDK6 complex hetero-dymerization promotes cell cycle progression by sequestering the p21^CIP1^ and p27^KIP1^ inhibitors of the cyclin E/CDK2 complex in late G1 and inhibiting negative checkpoints regulators such as Rb through phosphorylation, (see Figure 1).

In light of the critical role of the pathway cyclin D/CDK4 and CDK6/Rb in cell cycle progression, one can easily argue the relevance of its deregulation in cancer cells [18]. The presence of defective cyclin D, CDK4 and CDK 6 proteins, or mutations in their genes can promote both tumor initiation and progression [5,11,19,20,21].

CDK4 gene amplification has been detected in 50% of glioblastomas (GB) [22] while CDK4 activating point mutation in R24C has been commonly reported in melanoma patients with primary resistance to the INK4 family of CDKIs [23]. Similarly, *CCND1* over-expression has been reported in the carcinoma of the liver and CCND1 autophagy-dependent degradation leads to RB-driven inhibition of cell proliferation and tumor progression and subsequent suppression of the E2F transcriptional activity necessary for entering S phase [19]. Moreover in a xenograft model of nasopharynx carcinoma, other authors have shown that CDK4-6 inhibition with palpociclib exerts significant antitumor activity halting the cell cycle in G1 [24].

Altogether, these findings support the hypothesis that tumor cells may become functionally “addicted” to specific CDK alterations, thus making CDK inhibition a promising target for a new generation of anti-neoplastic agents.

Back in the early 1990s, the first CCDK2 inhibitor flavone L86-8275 blazed the trail toward the use of cell cycle inhibitors as a new ammunition against cancer [25]. More recently, a new generation of *CCND1*/CDK4 and CDK6 inhibitors, including alvocidib, palbociclib, ribociclib, and abemaciclib, demonstrated a relevant therapeutic activity in ESR+/HER2- advanced and metastatic breast cancer and is now a standard treatment in combination with aromatase inhibitors (AI) (anastrozole, letrozole, examestane), selective estrogen receptor (ESR) modulators (SERM: tamoxifen), and selective ESR down-regulators (SERD: fulvestrant).

Although several mechanisms have been evoked to explain the occurrence of primary and secondary resistance to this class of drugs, most of them involve *CCND1*/CDK 4 and CDK6/INK4/RB pathway [12] and highlight the critical role of Rb [26].

### 2.2. Ionizing Radiation and the Cyclin-D1/CDK4 and CDK6 Pharmacological Inhibition: Preclinical Evidence

From more than half a century [27], there has been a large consensus on the better response to ionizing radiation (IR) when cells are in G2/M rather than in G1/S [28,29]. Likewise, it is broadly agreed that IR determines a redistribution throughout the cell cycle; the repair of sub-lethal and potentially lethal DNA damage occur after IR; an indirect effect of IR is due to water radiolysis and reactive oxygen species (ROS); and IR induces endothelial activation and dysfunction in tissues.

Other evidence shows that cells treated with IR can repair damaged DNA or, on the contrary, undergo either programmed cell death pathways (PCD, e.g., apoptosis, autophagy) or necrosis [27]. An efficient cross-talk between the PCD pathways and necrosis has also been shown, which is not necessarily related to specific cell cycle phases [30]. Clear evidence does exist that tumor cells exposed to CCND1/CDK4 and CDK6 complex inhibitor increase their sensitivity to IR thanks to the induction of sustained DNA double-strand breaks (DSB) [31].

Although the precise mechanisms of these events have not been completely elucidated, several pre-clinical models provide the scientific rationale to investigate the combination pharmacological CDKIs [2,24,32,33,34,35,36] with IR [37] and suggest that it could be a paramount new avenue to achieve a better clinical performance of this class of drugs.

Consistently, other authors have provided evidence that in an orthotopic mouse glioblastoma model, combined use of palbociclib and IR achieved longer survival than palbociclib alone [2] and that in a xenograft cervical cancer model, two CDKIs, AT7519 and SNS-032, synergized with IR to induce apoptosis, senescence, and cytostasis [32].

These results were partially confirmed in clinical trials for brain cancer, when the use of CDKIs in combination with RT or temozolomyde or other anticancer agents showed promising results in terms of PFS and OS [38]. Similarly, the efficacy of CDKIs in combination with IR was also preliminarily tested in patients with lung [36], prostate, cervical, and breast cancer, suggesting a broad activity of this combination across these tumor types [38,39]. Notably, multiple pre-clinical studies have demonstrated the ability of CDKIs to promote apoptosis in tumor tissues exposed to IR, presumably by engaging p53 independent mechanisms

Interestingly, these studies also revealed the ability of CDKI to protect normal tissues by IR as well as chemotherapy-mediated damage by reducing the frequency of treatment associated intestinal injuries and hematological toxicity [40,41]. This phenomenon seems to be related to CDKIs’ ability to induce selective pharmacological quiescence of mucosal as well as hematopoietic cell precursors [42]. Several other similar points including the role of CDKIs in mitigating IR-induced neo-vasculogenesis, are certainly worthy of more future investigations [39].

Nevertheless, the optimal combined regimens with CDK4 and CDK6 inhibitors and RT, i.e., timing, dosage, treatment technique, and fractionation schedule, are still far from being identified.

### 2.3. Autophagy and the Cell Cycle

Autophagy is an evolutionary conserved process, aimed to maintain cellular homeostasis in response to stressful conditions (e.g., starvation, growth factors deprivation, excess of unfolded proteins, infection, gene mutation, etc., and altered signal pathways of cancer initiation and progression), and it can be also involved in type 2-PCD or autophagy-related cell death (APCD) in particular stressful situations [43].

Chaperone-mediated autophagy (CMA), micro-autophagy (or endosomal micro-autophagy (eMI), and macro-autophagy [44] are the three main process types. Among these, macro-autophagy, or simply “autophagy”, can be “basal” to degrade and recycle cytosolic components, as damaged macromolecules and organelles, to reconstitute the normal intracellular environment, or “induced” by external stress (e.g., starvation) to refill the cell aminoacidic pool for new proteins synthesis and/or energetic needs. Moreover, autophagy exerts pro-survival and anti-aging roles and is involved in tumor suppression and antigen presentation to immune effectors [45] and can contribute to apoptosis and other APCD processes. The autophagic process is divided into four different phases that are characterized by specific autophagy-related proteins (Atg) [46]: (1) initiation, (2) nucleation, (3) elongation and maturation, and (4) transport and fusion with lysosomes [47]. Briefly, the initial induction of the phagophore requires the formation of the Ulk1,2/Atg13/Atg101/FIP200 complex, whose activity is negatively or positively regulated by mTORC, depending on nutrients’ availability. The first phase depends on the formation of an autophagy-regulating PI3K complex (PI3K complex I) (Vps34/Beclin-1/Atg6/Atg14/p150/Vps15), that in turn is negatively regulated (nucleation) by the antiapoptotic proteins Bcl-2 or Bcl-xL that sequesters Beclin-1, thus inhibiting autophagosome’s formation. Subsequently, the third phase of the process requires the formation of the PI3K complex II (Atg12/Atg5/Atg16L1 complex) for the elongation of the pre-autophagosome membrane and the formation of the active form of the microtubule-associated protein 1/light chain 3 (LC3), designated as LC3-II, which is presently considered as the most reliable marker of autophagy. Finally, in the fourth phase of the process, mature autophagosomes fuse with endosomes, thus generating the amphysomes where the ultimate protein degradation occurs [48].

So far, no clear elucidation of the mechanisms underlying the cross-talk between autophagy and cell cycle has been achieved. CDKs-dependent cell progression can be related to autophagy inhibition, hence DNA damage or other inhibitory signals may be involved in inducing autophagy via CDKIs inhibition [49].

Many authors show a pivotal role of INK-4 and the Cip/Kip CDKIs families and, consequently, for Rb/E2F pathway in inducing autophagy [50].

p16, p21, and p27 Cip/Kip CDKIs have been strictly correlated to starvation-induced autophagy and cell cycle arrest. In particular, the cyclin-binding region of p27 appears to be essential in inducing autophagy and the depletion of CDK4 and CDK6 can reproduce, at least in part, this process. p16 overexpression has also been related to autophagy induction by promoting the Rb pathway. Moreover, some autophagy-promoting genes, including ULK and LC3, are the ultimate targets of the E2F transcription activity, and in turn, E2F factors were shown to bind the promoter site of Beclin-1. Therefore, the relationship between the Rb/E2F pathway and autophagy, and the possible correlation between autophagy and cell-cycle arrest, are characterized by a great complexity that explains some contrasting evidence due to multiple redundant pathways including, eight different known members within the E2F family with opposite regulative gene expression [51]. Polager et al. have provided evidence that the intensity of Rb/E2F pathway activation influences the autophagic response because a declined E2F1 expression hampers the Rb-driven DNA damage-induced autophagy. In the human bone osteosarcoma cell line U2OS [52], the presence of functional E2F1 transcription factors Rb pathway activation promotes autophagy inducing the expression of damage-regulated autophagy modulators genes encoding for LC3, Atg1, Atg5, and Dram. An excessive Rb activity or a defective E2F1 causes autophagy. In contrast, an excess in E2F1 activity antagonizes Rb-induced autophagy and promotes apoptosis [53]. This is consistent with the hypothesis that Bcl-2 might be a target for suppression of autophagy by binding Beclin-1.

It is noteworthy that Zheng et al. [54] reported several experimental conditions where both induction and suppression of autophagy by cell-cycle proteins (CDKs and CDKIs) were attributable to different stimuli and conditions. Similarly, recent evidence showed that chemical- or siRNA-based CDK4 and CDK6 inhibition can induce autophagy in multiple cancer cell lines [54,55]. In particular, treatment of myeloma cell lines with CDK4 and CDK6 inhibitor, abemaciclib, leads cells to accumulate in G0/G1, and exerts cytocidal activity accompanied by a dose-dependent autophagic vacuolization [56].

In addition, it has been demonstrated that autophagy itself can regulate the cell cycle. This two-way activity depends on the cell and TME conditions and may either lead cancer cells to cell death or may promote their survival [50,57]. The regulation of these events has been attributed to an autophagy-induced degradation of the cell cycle proteins, partially depending on the ubiquitin-proteasomal system (UPS). In particular, G1 arrest in breast cancer [58] and hepatocarcinoma cells [6] seems to be related to *CCND1* autophagy-induced degradation. The role of autophagy on *CCND1* was also supported by our group’s results obtained from surgical samples of resected GB relapsing after postoperative RT with concomitant and adjuvant temozolomide [59] (the gold standard for GB patients). We showed the co-localization of *CCND1* with the key autophagy protein Beclin-1 and a possible role for autophagy in the control of cell cycle. One more promising strategy to overcome GB resistance is the use of PI3K/AKT/mTOR pathway inhibitors, which are demonstrated to increase IR-sensitivity and autophagy-related cell death in GB cell lines [60]. Altogether, these results prompted us to hypothesize that autophagy-related *CCND1* degradation, mTOR inhibition, and autophagy activation could be a novel valuable therapeutic strategy to treat human tumors

### 2.4. The PI3K/AKT/mTOR Pathway in Endocrine Resistant ESR1+/HER2− Breast Cancer, and in other Neoplasms

The PI3K/AKT/mTOR signaling cascade is involved in escape pathways from hormonal therapy (HT) in breast cancer and may be independent of ER activation [61]. Because this signaling impacts on the cell cycle, cyclin-D1/CDK4 and CDK6 function, autophagy and IR are involved in many human neoplasms. It may be due to mutations or amplifications of genes, such as the encoding *PI3KCA*, or PI3K downstream components (PTEN, acting as a suppressor; AKT; mTOR). The relevance of PI3K signaling in particular has been shown in pre-clinical models of prostate and lung cancer, melanoma, leukemia, hematological malignancies, etc.

ER stimulation receptor tyrosine kinase (RTK) activation by growth factors and extracellular matrix components can activate PI3K, whereas RTK activation is cross-linked with the MAPK pathway [62]. Activation by PI3K is necessary for AKT through the phosphorylation of Thr-308 and Ser-473 by the phosphoinositide-dependent kinases (PDK)-1 and -2. Activated AKT, in turn, regulates downstream effectors including mammalian target of rapamycin (mTOR), often deregulated in human cancer.

It is noteworthy that AKT can be activated independently from the PI3K signaling, by IR-induced DNA double-strand break [63], and that PI3K/AKT/mTOR pathway is crucial for the functional interaction between *CCND1* and CDK4 and CDK6 [64].

## 3. The State of the Art in the Clinical Domain

### 3.1. Clinical Trials of Cyclin-D1/CDK4 and CDK6 Inhibition in Breast Cancer

Despite the wide experimental preclinical work on cyclin-D1/CDK4 and CDK6 inhibitors in several neoplasms, these agents have been mostly clinically tested only for advanced and metastatic malignancies and just recently have been approved as a standard first-line strategy in HR-positive, HER2 negative breast cancer, in various combinations with AIs, SERMs, or SERDs.

In a phase II clinical trial (PALOMA-1), the addition of palbociclib to letrozole significantly increased the median progression-free survival (PFS) from 10.2 months to 20.2 months [15] vs. letrozole alone for post-menopausal women with previously untreated ESR1+/HER2− advanced breast cancer. Palbociclib received accelerated approval by the US Food and Drug Administration (FDA) in February 2015 [65]. The placebo-randomized, double-blind, controlled phase III trial (PALOMA-3) compared treatment with palbociclib and fulvestrant to fulvestrant plus placebo in women with ESR1+/HER2− metastatic breast cancer that had relapsed or progressed on prior hormone therapy, including a substantial portion of patients (33%) with prior chemotherapy for metastatic disease. The interim analysis of this study demonstrated a significantly improved median PFS (9.5 months versus 4.6 months, respectively) [12,66]. Palbociclib is presently under further investigation in over 50 trials in related clinical settings.

Ribociclib is a selective strong inhibitor of CDK4 and CDK6 [13], blocks Rb phosphorylation, and causes cell cycle arrest of Rb-positive tumor cells, similarly to palbociclib [14]. The MONALEESA-2 trial demonstrated that ribociclib plus letrozole significantly improved progression-free survival, compared with placebo plus letrozole as a first-line therapy in postmenopausal patients with HR-positive, HER2-negative advanced breast cancer [15].

Ongoing trials are demonstrating significant progression-free survival improvements with ribociclib in different settings [67,68]. However, both palpociclib and ribociclib were characterized by a not-negligible systemic toxicity [31].

Abemaciclib was tested in a phase I trial, showing efficacy against non-small cell lung cancer, melanoma, and mantle cell lymphoma, in addition to breast cancer [69]. In fact, other than its high affinity for CCND1/CDK4, it also inhibits cyclin-B/CDK1 (related to G2 arrest) and cyclin E-A/CDK2 (related to palpociclib resistance). Differently from the cytostatic effect alpociclib and ribociclib, abemaciclib achieved a durable cytotoxic effect in Rb-negative cells [70]. Moreover, in a first line randomized phase III double-blind study (MONARCH 3) for ESR1+/HER2− advanced breast cancer, abemaciclib presented a better hematological tolerability compared to palpociclib and ribociclib, and achieved impressive results in terms of PFS and objective response rate (ORR) [71]. A synthesis of these clinical trials is reported in Table 1**.**

A metanalysis demonstrated that three CDK4 and CDK6 inhibitors have similar efficacy when associated with AIs, and are superior to either fulvestrant or AI monotherapy independently of any stratification criteria, thus supporting AI plus CDK4 and CDK6 inhibition as the best approach in ESR1+/HER2− metastatic breast cancer [65,66]. Li et al., in another meta-analysis, included the most prominent nine previous prospective trials and 5043 patients [72]. Adding CDK4 and CDK6 inhibitors to endocrine therapy achieved a highly significant benefit for OS, compared to ET alone. Moreover, significant gain in OS also resulted in second-line treatment in patients affected by visceral metastases. However, the combined CDK4 and CDK6 inhibitor/ET treatment was characterized by a heavier G3-4 toxicity that was less pronounced for abemaciclib. Nevertheless, it should be remembered that 10% of the overall ESR1+/HER2− breast cancer patients show primary or secondary resistance to this combined therapeutic approach [6].

All the chemical structures of the molecules are presented in Figure 2.

### 3.2. Real-Life Clinical Reports on Cyclin-D1/CDK4 and CDK6 Inhibitors, Including Association with Radiotherapy

Delivering unplanned RT (mainly with palliative/symptomatic aims) in advanced breast cancer patients on Cyclin-D1/CDK4 and CDK6 inhibitors in prospective trials has produced empirically based treatment not supported by an appropriate rationale and consolidated approach, e g., in terms of RT intent, dose, and scheduling. However, several observational studies provided clearly suggest the lack of additive adverse events when palbociclib or ribociclib are combined with letrozole and concomitant palliative RT [73,74,75]. Additionally, a retrospective single-institution analysis suggested that 15 out a cohort of 42 pts with ESR1+ breast cancer with brain metastases showed promising results using CDK4 and CDK6 inhibitors palbociclib or abemaciclib for 6 months combined with stereotactic radiation [71] and only 5% of the patients developed radiation necrosis. However, only a slight improvement could be shown in survival. However, early radiation toxicities, including esophagitis and severe dermatitis have been reported in a breast cancer patient on palpociclib and palliative RT on cervical lymph nodes [72]. Similar studies evaluating CDK inhibitors and RT are presently ongoing in patients with head and neck, brain, breast, and pancreatic cancer as reported in Table 2.

## 4. Molecular Grounds for Future Developments

### 4.1. CCND1 as a Target of Ionizing Radiation in the CCND1-CDK4, and CDK6 Complex

Only the CCND1-CDK4 and CDK6 complex has raised interest as a possible actionable target, whereas, in contrast, despite its prognostic relevance in cancer [76], the regulatory member CCND1 per se has received less attention. As previously quoted, palpociclib induces autophagy in cancer, targeting CDK4 and CDK6 [54,55], via the LKB1 and Ser325 phosphorylation resulting in a CCND1-mediated, reduced activation of AMPK [77]. The *CCND1* gene encoding CCND1 is activated in B-cell lymphomas and overexpressed in many other human cancers (esophageal, lung, breast, bladder carcinomas, etc.) where it acts as a proto-oncogene. Therefore, the CCND1 is a potential pivotal actionable target for cell cycle arrest. In normal cells, CCND1 has a very short life due to UPS degradation whereas in cancer cells, inhibition of turnover and increased levels of CCND1 were shown. Conversely, the knockdown of the activity of a specific deubiquitinase for CCND1 was shown, inducing growth suppression in cancer cells addicted to its expression [78]. Ubiquitination of CCND1 is achievable in response to IR [79] and, in cancer cells, IR or other DNA-damaging agents may activate selective autophagy [54]. Autophagy shares common molecular mechanisms with UPS, with which it is intertwined in a close and complex relationship [80]. As described by these authors, ubiquitin-specific recognition can have a pivotal role for CCND1 autophagy degradation. Selective autophagy requires some receptor proteins necessary for the process, such as p62 (also known as sequestosome 1 or SQSTM1), NBR1 (neighbor of the *BRCA 1* gene), CALCOCO2 (calcium binding and coiled-coil domain 2), and optineurin. p62 (the most relevant receptor protein for the subject dealt with here) is a component of several transduction signal cascades, such as Keap-Nrf2, and essential for UPS degradation.

Possible future strategies aimed at enhancing the IR-autophagic degradation of CCND1 in combination with drugs targeting the CCND1-CDK4 and CDK6 complex for G1/S arrest could overcome the IR-resistance of G1 and S phases of cell cycle

### 4.2. Ionizing Radiation, Autophagy Enhancement, and CCND1

Ionizing radiation can directly induce autophagy; although the mechanism underlying this effect has not been fully elucidated so far, two main mechanisms have been evoked: (1) similarly to rapamycin, IR reverts auto-phosphorylation of mTOR (e.g., in MCF-7 breast cancer cell culture) leading to the formation of autophagic vacuoles, segregation of cytoplasmic materials, and appearance of LC3 positivity and eventually mitochondrial hyperpolarization and decreased cell survival [81].

(2) Radiation causes a protective pathway via endoplasmic reticulum (ER) stress, accumulation of ER misfolded proteins (unfolded protein response—UPR), and autophagy. Therefore, UPR pathway and autophagy are upregulated in some cancer types as an escape strategy leading to IR-activated autophagy responsible for resistance to RT [82,83,84]. Nonetheless, autophagy is always depictable as a “double-edged sword”. As previously reviewed [80], most of the literature dedicated to this subject is focused on drug manipulation of autophagy (i.e., either induction or inhibition) for improving IR-sensitivity in cancer cells, more than to IR in conditioning, per se, autophagy towards cell death.

Emphasis should be placed in autophagy during the cell cycle, as different studies have investigated the autophagy activation in the different phases of the cell cycle [81,82]. In contrast, other evidence suggests a protective suppression of autophagy during mitosis aimed at preventing the inappropriate inactivation of molecular species and organelles, functional for cellular replication, or even aggregated chromosomes [50]. IR-based autophagy induced during the G1 and S counteracts the accelerated cell cycle in cancer and lays the groundwork for combined therapies combined with IR. It has been shown that *CCND1* suppresses autophagy in mammary epithelium [85] whereas *CCND1* inhibits autophagy by activating AMPK in breast cancer cells [77]. Further, in 2016, our group published a study in recurrent GB patients that prompted us to hypothesize a role for the IR-temozolomide treatment in inducing *CCND1* autophagic degradation in Beclin-1-positive GBs [59]. Consistently, another group demonstrated the inverse correlation between low autophagic activity and high *CCND1* expression that were also related to poorer prognosis [4]. In human and murine models of hepatocellular carcinomas (HCC), induction of autophagy led to *CCND1* ubiquitination and its recruitment into autophagosomes via SQSTM1, formation of autophago-lysosomes, and degradation with suppression of tumor growth. Further, Chen et al. in 2019 [58] showed that autophagy itself causes *CCND1* autophagic degradation in breast cancer cell lines with subsequent G1 cell cycle arrest underpinning the hypothesis of cell cycle modulation by CCND1 via mTORC1 signaling inhibition. Finally, we proposed in 2020 an overview of the role of autophagy, and namely its pivotal Beclin-1 protein, in *CCND1* degradation, as a possibly actionable mechanism for G1/S cell cycle arrest in cancer (Figure 3) [86].

It is still unclear whether autophagy could represent a true per se pro-death process, or can simply enhance the IR and chemotherapy sensitivity of cancer cells. The impact of autophagy in a cell death progress is cautiously named “autophagy-related cell death”. This definition is underpinned by the clinical observation that a high tissue expression of cytoplasmic Beclin-1 was related to better prognosis in patients with malignant glioma on temozolomide chemotherapy and radiotherapy [87]. Notwithstanding, one should also take account of the tumor suppression of the autophagy-independent functions of Beclin-1 [88].

## 5. Concluding Remarks, with Special Regard to Immunotherapy Developments

Although the efficacy of combining HT and CDK4 and CDK6 inhibitor in patients with ESR1+/HER2− breast cancer, the promising benefit of CDK4 and CDK6 drug inhibitors in other neoplasms has not been extensively tested.

Attempts to enhance the G1/S arrest action are currently under clinical investigation, e.g., based on a concurrent PI3K-pathway inhibition for breast cancer (triple therapy), or a sequential use of these agents in cases bearing *PIK3CA* mutations [8]. Of note, PALOMA trials showed that *CCND1* amplification and overexpression did not correlate with reduced therapeutic results. We believe this latter observation does not rule out *CCND1* being an actionable target. Notably, there is evidence that autophagy induction by IR and/or drug manipulation can induce *CCND1* degradation not only in breast cancer [58], but also in hepatocellular carcinoma (HCC) [6] and GB cells [59,86]. Other than with IR in vitro, autophagy can be induced by alkylating drugs (e.g., temozolomide) and mTOR inhibitors (e.g., rapamycin and rapalogs), although their combination with CDK4 and CDK6 inhibitors raises safety concerns. On the other hand, in vitro experiments have shown that two pro-autophagy drugs, amiodarone (in HCC; [6]) and trehalose (in melanoma), are safer or even show no toxicity at all, respectively. Moreover, more recently, selective and less toxic mTOR inhibitors showed reduced impact rapamycin or rapalogs on glucose homeostasis, lipid metabolism, and immune response [89].

A high autophagy rate degrades cell cycle regulators, including cyclin-induced senescence [90]. Cancer cells secrete inflammatory cytokines, chemokines, and growth factors (collectively designated as senescence-associated secretory phenotype, or SASP) [31] and CDK inhibitors and IR may contribute to this process. SASP may also recruit immunity against the senescent cells, whereas other combined actions by CDK inhibitors and IR, i.e., enhancement of the MHC class I expression, activation of dendritic cells, increased antigen cross-presentation, and reduced the proliferation of Tregs. These effects do contribute not only to local but also systemic cancer-suppressive response (abscopal effect by IR) [91,92].

It is worth mentioning that, according to nutrient availability, microenvironment stress, pathogenic conditions, and a responsive immune system, autophagy plays neutral, tumor-suppressive, or tumor-promoting roles [93]. Different treatment responses may, therefore, be in part due to factors that influence the immune system (thus, autophagy) other than IR, such as genetic background, dietary habits, and microbiome. Accounting for these factors and their impact on the treatment response, the research framework of molecular pathological epidemiology (MPE) has already been demonstrated to offer a valid interdisciplinary integration when it was adopted to study breast, lung, prostate, and colorectal cancers. The MPE approach can be applied, for instance, to assess the influence of exogenous and endogenous factors on carcinogenic processes by gauging the immune response to a tumor, filling the research gap between tumor immunology and epidemiology (immune-MPE) [94].

MPE research must work with both in vivo and in vitro experimental research to improve our understanding of disease pathogenesis, given the complexities of human diseases, to fulfill an integrative scientific approach [95].

Moreover, platform database analysis addressed *CCND1* amplification as a potential surrogate biomarker of efficacy and even target for immunotherapy [96]. *CCND1* amplification was associated with the exclusion of cytotoxic cells, T CD8+ and B+ T cells and dendritic cells (DCs) in the tumor microenvironment (TME). These studies also suggest a possible correlation between *CCND1* amplification and immunosuppressive features as the epithelial–mesenchymal transition (EMT), transforming growth factor (TGF)-b, KRAS and PI3K/AKT/mTOR signaling, p53 pathway, and hypoxia signaling.

In light of the data available, *CCND1* is an attractive, multifaceted target for cancer therapy and is worth extensive study to explore its impact as a novel actionable target and when combined with IR.

Setting up proper patient stratification and study design will allow us to understand how to unleash this new ammunition against cancer.

## Figures and Tables

**Figure 1 ijms-22-08391-f001:**
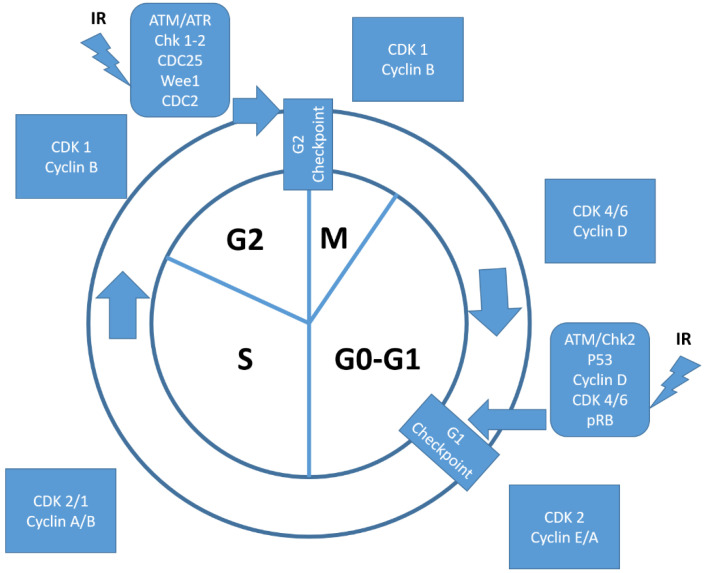
Schematic overview of cell cycle regulation, with an emphasis on radiotherapy-induced pathways and CDK/cyclin regulation. In M phase and in G2 resting phase, cancer cells are respectively very sensitive and moderately sensitive to radiation injury, whereas in G1 phase and in S phase, cancer cells are moderately resistant to radiation injury. Irradiation induces G1 and G2 cell cycle checkpoint activation and DNA repair. Most cancer cells are defective in G1 checkpoint, commonly due to the mutations/alterations of the key regulators of the G1 checkpoint, but contain a functional G2 checkpoint.

**Figure 2 ijms-22-08391-f002:**
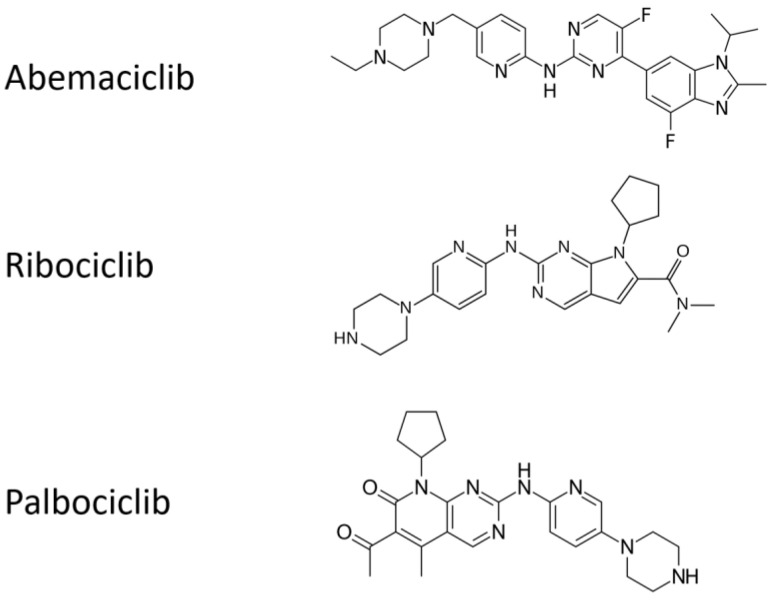
Chemical structures of cyclin inhibitors that are currently approved.

**Figure 3 ijms-22-08391-f003:**
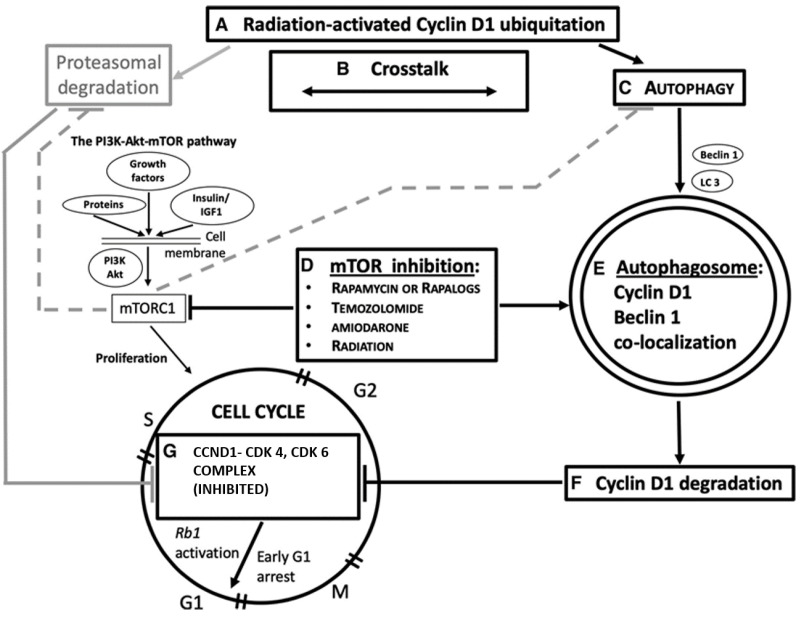
A simplified overview of the role of autophagy and Beclin-1 in *CCND1* degradation, as a possible mechanism actionable for G 1/S cell cycle arrest in cancer. Radiation and chemotherapy are also considered in this domain. Some steps of the pathway are indicated, as shown in different tumors. A: Ubiquitation in *CCND1* levels is achievable by gamma irradiation [3]. B: Ubiquitin–proteasome system and autophagy functionally cooperate with one another for protein homeostasis [1]. C: Autophagy is induced in response to environmental stress resulting in abnormal or degraded proteins [5]. Beclin-1 (encoded by the *BECN1* gene) is an autophagy protein, which has a critical role in the autophagy machinery. LC3 is a protein of the late autophagy pathway, necessary for autophagosome, biogenesis, and is used as an autophagy marker. D: Mammalian target of rapamycin (mTOR) inhibitors (everolimus [2], rapamycin [6,11], amiodarone [11]) activate autophagy, also concurrently with radiation or temozolomide chemotherapy [6]. E: The autophagosome engulfs degraded proteins [5], subsequently digested via its fusion with lysosomes. *CCND1* colocalizes with Beclin-1 in the cytoplasm [7] after cancer therapy and this suggests its autophagy degradation. F and G: *CCND1* degradation deactivates the formation of the CCND1/CDK4 and CDK6 complex, thus activating the Rb1-dependent onco-suppressive pathway [10] and hampering cell cycle progression [2,6]. In gray: an alternative pathway of *CCND1* degradation is through the ubiquitin–proteasome mechanism (UPD). Active mTOR complex 1 (mTORC1) inhibits both UPD and autophagy (dashed lines). PI3K, phosphatidylinositol 3-kinase. Image copyright © 2021, The American Physiological Society [86].

**Table 1 ijms-22-08391-t001:** Synthesis of the registrative clinical trials of cyclin inhibitors in metastatic breast cancer.

Scheme.	INTERVENTION	PATIENT CHARACTERISTICS	MEDIAN PFS(Months)	Clinical Gain/Approval
PALOMA-1	Palbociclib + letrozole vs. letrozole alone	Post-menopausal women with untreated ER+/HER2−advanced breast cancer	20.2 vs. 10.2	Significant gain in terms of median PFS (10 Months). FDA approval in 2015
PALOMA-2	Palbociclib + letrozole vs. letrozole alone	ESR1+/HER2−advanced breast cancer	27.6 vs. 14.5	Delayed ChT: 40.4 vs. 29.9 months
PALOMA-3	Palbociclib + fulvestrant vs. placebo + fulvestrant	ESR1+/HER2− metastatic breast cancer after hormone therapy (30% received prior ChT)	9.5 (11.2) vs. 4.6	Significant gain in terms of median PFS
MONALEESA-2	Ribociclib + letrozole vs. letrozole alone	Postmenopausal women with ESR1+/HER2 advanced breast cancer	Not reached vs. 14.7	FDA approval in 2017
MONARCH-3	Abemaciclib + aromatase inhib. vs. placebo + aromatase inhib.	Postmenopausal women with ESR1+/HER2 locoregionally recurrent or metastatic breast cancer with no prior systemic therapy	28.1 vs. 14.7	Significantly prolonged PFSFDA approval in 2018

**Table 2 ijms-22-08391-t002:** Ongoing clinical trials evaluating combination of CDK inhibitors and RT in different diseases.

NCT Number	Study Type	Cancer Type	Trial Title	Sponsors and Collaborators
NCT02290145	Interventional, Phase II	Squamous cell carcinoma of mouth	CCND1 Based TPF Induction Chemotherapy for Oral Squamous Cell Carcinoma Patients at Clinical N2 Stage	•Ninth People’s Hospital, Shanghai Jiao Tong University School of Medicine, Shanghai, Shanghai, China
NCT04585724	Interventional, Phase I	Metastatic breast carcinoma	Stereotactic Radiosurgery with Abemaciclib, Ribociclib, or Palbociclib in Treating Patients with Hormone Receptor Positive Breast Cancer with Brain Metastases	•Grady Health System, Atlanta, GA, United States
NCT03024489	Interventional, Phase II	Head and neck cancer	Palbociclib with Cetuximab and IMRT for Locally Advanced Squamous Cell Carcinoma	•Faculty of Medicine, Ramathibodi Hospital, Bangkok, Thailand
NCT02607124	Interventional, Phase II	High grade glioma	A Phase I/II Study of Ribociclib, a CDK4 and CDK6 Inhibitor, Following Radiation Therapy	•Cincinnati Children’s Hospital Medical Center, Cincinnati, OH, United States
NCT04298983	Interventional, Phase II	Prostate cancer	Abemaciclib in Combination with Androgen Deprivation Therapy for Locally Advanced Prostate Cancer	University of Alabama at Birmingham
NCT04923542	Interventional, Phase II	Brain metastases from breast cancer	Stereotactic Radiation and Abemaciclib in the Management of ESR1+/HER2- Breast Cancer Brain Metastases	•H. Lee Moffitt Cancer Centerand Research Institute
NCT04220892	Interventional, Phase I	High grade glioma	Pilot Study of Pembrolizumab Combined with Pemetrexed or Abemaciclib for High Grade Glioma	•Jose Carrillo •Eli Lilly and Company •John Wayne Cancer Institute
NCT03355794	Interventional, Phase I	Cerebral glioma	A Study of Ribociclib and Everolimus Following Radiation Therapy in Children with Newly Diagnosed Non-biopsied Diffuse Pontine Gliomas (DIPG) and RB+ Biopsied DIPG and High Grade Gliomas (HGG)	Children’s Hospital Medical Center, Cincinnati, OH, United States •Novartis
NCT03691493	Interventional, Phase II	Bone metastases from breast cancer	Radiation Therapy, Palbociclib, and Hormone Therapy in Treating Breast Cancer Patients with Bone Metastasis	•Emory University •Pfizer
NCT03870919	Interventional, Phase I	Breast cancer stage IV	Locoregional Treatment and Palbociclib in de Novo, Treatment Naive, Stage IV ESR1+, HER2- Breast Cancer Patients	•UNICANCER •Pfizer
NCT03024489	Interventional, Phase II	Head and neck cancer	Palbociclib with Cetuximab and IMRT for Locally Advanced Squamous Cell Carcinoma	•Mahidol University
NCT04563507	Interventional, Phase II	Breast cancer stage IV	Combined Immunotherapies in Metastatic ESR1+ Breast Cancer	•Weill Medical College of Cornell University
NCT03389477	Interventional, Phase II	Head and neck cancer	Los Tres Paso: Neoadjuvant Palbociclib Monotherapy, Concurrent Chemoradiation Therapy, Adjuvant Palbociclib Monotherapy in Patients with p16INK4a Negative, HPV- Unrelated Head and Neck Squamous Cell Carcinoma	•Washington University School of Medicine •Pfizer
NCT04605562	Interventional, Phase II	Nasopharyngeal carcinoma	Umbrella Biomarker-Guided Therapy in NPC	•Sun Yat-sen University
NCT02624973	Interventional, Phase II	Breast cancer	PErsonalized TREatment of High-risk MAmmary Cancer—the PETREMAC Trial	•Haukeland University Hospital •Helse Vest •Pfizer •AstraZeneca

## Data Availability

Not applicable.

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
