# Peer review of "CDK4, CDK6/cyclin-D1 Complex Inhibition and Radiotherapy for Cancer Control: A Role for Autophagy"

_ijms, 2021, doi:10.3390/ijms22168391_

Round 1

Reviewer 1 Report

The article describes the expanding clinical application of CDK4/6 inhibiting drugs in the managements of breast cancer which has raised interest in testing these drugs in other neoplasms. The potential of combining these drugs with other therapeutic approaches seems an interesting work-ground to explore. The review discuss the recent literature on the field in order to develop a combination strategy including Cyclin-D1/CDK 4-6 inhibitors, radiotherapy and/or other anticancer agents targeting G1-S phase cell cycle transition.

Chemical structures and physico-chemical of the drugs in question should be presented. 

Author Response

We thank the Editors and the Reviewers for their efforts reading and evaluating our manuscript. We believe that thank to your help our manuscript has largely improved.

Reviewer 1:

Point 1: Chemical structures and physico-chemical of the drugs in question should be presented. 

ANSWER

We have provided a figure (Figure 2) with the chemical structure of the drugs currently approved as cyclin inhibitors.

Reviewer 2 Report

The authors wrote a quite interesting review on CDK inhibition and radiotherapy. This is generally of interest. Following things should be addressed and the authors should improve the paper.

Differences in treatment response may be in part due to lifestyle factors. There are many environmental, dietary, and lifestyle factors that influence immune system and pathogenic mechanisms. The authors should discuss those. There are also influences of germline genetic variations on both immune system and cancer progression. Gene-by-environment interactions should be discussed.

In these contexts, research on dietary / lifestyle factors, immunity, and personalized molecular biomarkers in tumor is needed for prevention and treatment research. The authors should discuss molecular pathological epidemiology research that can investigate those factors in relation to microbiome, molecular pathologies, immunity, and clinical outcomes. Molecular pathological epidemiology research can be a promising direction. Strengths and challenges of molecular pathological epidemiology (J Gastroenterology 2017, Ann Rev Pathol 2019, etc.) should be discussed.

The authors wrote CDK4,6, CDK4/6, CDK 4 / 6, CDK4-6, CDK 4-6, all over the place. CDK4 and CDK6 should be spelled out completely as CDK4 and CDK6. Look at www.genenames.org. They should not have a comma, a slash, a space, a dash, or a hyphen. CDKs have better organized nomenclature system.

The authors should write CCND1 (cyclin D1) or CCND1. Look at www.genenames.org

The authors used ER for endoplasmic reticulum and ER for estrogen receptor. This is not acceptable. Write ESR1 (estrogen receptor 1). Look at www.genenames.org

Figures and tables should be revised. Do not use a hyphen, a slash, a comma or a space within any protein name.

Author Response

We thank the Editors and the Reviewers for their efforts reading and evaluating our manuscript. We believe that thank to your help our manuscript has largely improved.

Points raised

Reviewer 2:

Point 1: Differences in treatment response may be in part due to lifestyle factors. There are many environmental, dietary, and lifestyle factors that influence immune system and pathogenic mechanisms. The authors should discuss those. There are also influences of germline genetic variations on both immune system and cancer progression. Gene-by-environment interactions should be discussed. In these contexts, research on dietary / lifestyle factors, immunity, and personalized molecular biomarkers in tumor is needed for prevention and treatment research. The authors should discuss molecular pathological epidemiology research that can investigate those factors in relation to microbiome, molecular pathologies, immunity, and clinical outcomes. Molecular pathological epidemiology research can be a promising direction. Strengths and challenges of molecular pathological epidemiology (J Gastroenterology 2017, Ann Rev Pathol 2019, etc.) should be discussed.

Answer

We have modified the conclusions in the manuscript in order to address the interesting points suggested by the Reviewer.

 It's worth mentioning that, according to nutrient availability, microenvironment stress, pathogenic conditions, and a responsive immune system, autophagy plays neutral, tumorsuppressive, or tumor promoting roles. (Li et al. Molecular Cancer 2020, Autophagy and autophagy-related proteins in cancer) Different treatment responses may , therefore, in part due to factors that influence the immune system (thus, autophagy) other than IR, such as genetic background, dietary habits, and microbiome. Accounting for these factors and their impact on the treatment response, the research framework of molecular pathological epidemiology (MPE) already demonstrated to offer a valid interdisciplinary integration when it was adopted to study breast, lung, prostate, and colorectal cancers. The MPE approach can be applied, for instance, to assess the influence of exogenous and endogenous factors on carcinogenic processes by gauging the immune response to a tumor, filling the research gap between tumor immunology and epidemiology (immune-MPE). (Ogino S, Nowak JA, Hamada T, Phipps AI, Peters U, et al. 2018 Integrative analysis of exogenous, endogenous, tumour, and immune factors for precision medicine)

MPE research must work with both in vivo and in vitro experimental research to improve our understanding of disease pathogenesis, given the complexities of human diseases, to fulfill an integrative scientific approach. (Shuji Ogino, Jonathan A. Nowak, Tsuyoshi Hamada, Danny A. Milner Jr., and Reiko Nishihara “Insights into Pathogenic Interactions Among Environment, Host, and Tumor at the Crossroads of Molecular Pathology and Epidemiology”, Annu Rev Pathol. 2019)

Point 2: The authors wrote CDK4,6, CDK4/6, CDK 4 / 6, CDK4-6, CDK 4-6, all over the place. CDK4 and CDK6 should be spelled out completely as CDK4 and CDK6. Look at www.genenames.org. They should not have a comma, a slash, a space, a dash, or a hyphen. CDKs have better organized nomenclature system. The authors should write CCND1 (cyclin D1) or CCND1. Look at www.genenames.org. The authors used ER for endoplasmic reticulum and ER for estrogen receptor. This is not acceptable. Write ESR1 (estrogen receptor 1). Look at www.genenames.org Figures and tables should be revised. Do not use a hyphen, a slash, a comma or a space within any protein name.

Answer: we have modified the manuscript, as well the tables following the Reviewer suggestions.